# How does priority setting for resource allocation happen in commissioning dental services in a nationally led, regionally delivered system: a qualitative study using semistructured interviews with NHS England dental commissioners

Christopher Robert Vernazza,[1,2] Greig Taylor,[1,2] Cam Donaldson,[3] Joanne Gray,[4] Richard Holmes,[1] Katherine Carr,[1] Catherine Exley[5]

For numbered affiliations see end of article.

**Correspondence to**
Dr Christopher Robert Vernazza;
c.r.vernazza@ncl.ac.uk

## ABSTRACT

**Objectives** To understand approaches to priority setting for healthcare service resource allocation at an operational level in a nationally commissioned but regionally delivered service.

**Design** Qualitative study using semistructured interviews and a Framework analysis.

**Setting** National Health Service dentistry commissioning teams within subregional offices in England.

**Participants** All 31 individuals holding the relevant role (dental lead commissioner in subregional offices) were approached directly and from this 14 participants were recruited, with 12 interviews completed. Both male and female genders and all regions were represented in the final sample.

**Results** Three major themes arose. First, 'Methods of priority setting and barriers to explicit approaches' was a common theme, specifically identifying the main methods as: perpetuating historical allocations, pressure from politicians and clinicians and use of needs assessments while barriers were time and skill deficits, a lack of national guidance and an inflexible contracting arrangements stopping resource allocation. Second, 'Relationships with key stakeholders and advisors' were discussed, showing the important nature of relationships with clinical advisors but variation in the quality of these relationships was noted. Finally, 'Tensions between national and local responsibilities' were illustrated, where there was confusion about where power and autonomy lay.

**Conclusions** Commissioners recognised a need for resource allocation but relied on clinical advice and needs assessment in order to set priorities. More explicit priority setting was prevented by structure of the commissioning system and standard national contracts with providers. Further research is required to embed and simplify adoption of tools to aid priority setting.

## Strengths and limitations of this study

► The qualitative nature of this study allows a complex area to be explored in depth.
► The national sample adds value beyond the case studies, which predominate in this area.
► The operational rather than strategic level of those recruited offers an unusual insight into priority setting practice at this level.
► The specific area studied (dentistry) may be unusual and the variation in health systems globally may mean the insights offered are very specific.
► The sample size was limited, however, data saturation was reached.

## INTRODUCTION

In any healthcare system, those responsible for organising the provision of services (in England, commissioners) have difficult choices to make in that there are insufficient resources to provide all possible services.[1] Prioritisation for resource allocation must, therefore, be undertaken and this is often a complex task as healthcare systems have multiple competing objectives.[2] Although historically, taking England prior to 2012 as an example, prioritisation was undertaken implicitly using inefficient methods (such as perpetuating historical resource allocations or allocating resources based on pressure from clinicians, the public or policy-makers and politicians), there is an increasing recognition that more explicit, formalised approaches to priority setting would be useful.[3] These explicit approaches involve making decisions whereby options

for change (both investments and disinvestments) are assessed against each other on the basis of defined criteria, weighted for relative importance and considered also against the cost of each such option.

Many frameworks, methods and tools have been described for disinvestment (one aspect of resource allocation)[4] and several of the described frameworks actually deal with the wider task of priority setting for resource allocation, for example, multicriteria decision analysis (MCDA),[5] accountability for reasonableness (A4R),[6] Programme Budgeting Marginal Analysis (PBMA)[7] or socio-technical allocation of resources (STAR).[8] Tools have also been developed to operationalise these priority setting frameworks, for example, Public Health England's Prioritisation Framework (although this is specific to commissioning of public health interventions).[9]

Despite the recognised need for explicit priority setting and the development of tools, the extent of actual use of explicit mechanisms remains underwhelming. The majority of research into use of such mechanisms often describes academically led case studies, undertaken as part of research studies and offers very little evidence of ongoing use and adoption beyond the academic case study.[10] This case study-based approach also does not illuminate what those charged with resource allocation are actually doing in terms of priority setting in the absence of academic input. Four studies which looked at what was happening without or before academic involvement are relevant however and are described here. The first was a study looking at disinvestment decisions at regional level in Wales with senior managers.[11] This study found that although all of the regions planned to undertake explicit prioritisation exercises, none had actually done this yet and that it was difficult to identify disinvestments required to free up resources to allow nationally mandated investments to occur. The second study in England reported on the forms of priority setting undertaken at local level, finding that there was a large variation in practice and that epidemiological needs data were used by nearly all respondents compared with less than one-quarter using decision support tools.[3] The third example involved interviews with directors of finance in Scottish health boards, showing that none used explicit frameworks but that some had developed their own sets of criteria to judge decisions against.[12] All three of these studies, although undertaken at local levels, relied on responses from individuals at director level, who may not be involved in actual resource allocation decisions (as noted in the Welsh study). The fourth and final example looked at what was happening at decision-making level in a public hospital and community care group in Australia, involving a wide range of staff, including those at decision-making level. Although the focus of the study was disinvestments, a complex set of decision-making processes were found, but there were very few set criteria or explicit frameworks/processes for decision-making.[13]

This study, therefore, aimed to understand approaches to priority setting at decision-maker (rather than director) level.

## METHODS
### Context
The chosen exemplar area for the study is National Health Service (NHS) dentistry in England which currently incurs a spend of around £3.7 billion per year or 3.5% of the NHS budget.[14] In the NHS in England, the 'World Class Commissioning' programme set out expectations of commissioners and frameworks for commissioning. At the heart of this, is the commissioning cycle[15] which is split into three phases: strategic planning, procuring services and monitoring and evaluation. The phase most relevant to this study is strategic planning which is subdivided into three subphases: assessing need, reviewing service provision and setting priorities.

Unlike much of NHS commissioning in England, a national body (a commissioning board now known as NHS England) is responsible for commissioning of dentistry with part development to subregional teams.[14] The size of subregional teams varies but as an exemplar, Greater London is a region which has four subregions within it. Although the budget for dentistry in devolved to each region and it is established that subregional teams have the power to and responsibility for procuring dental services in their region, there is no formal agreement about where the power and responsibility for strategic decisions such as priority setting for allocating resources, or for other unusual situations (such as legal challenges to procurement decisions). It is worth noting here that at the time of the study, early preparations were being made for a much larger scale devolution of governmental power for one region (Greater Manchester) which would include total devolution of all health budgets and power.

In each subregional team, a lead commissioner for dentistry is responsible for purchasing all dental services required for their population from providers, which are, in the main, primary care dental practices with smaller amounts allocated to specialist providers in both primary and secondary care settings. The primary care contract is mainly based around a standardised national contract in which a certain number of units of dental activity (UDAs) are bought from providers for a negotiated price. The UDAs are generated by provision of three different bands of treatment generating 1, 3 and 12 UDAs depending on the type of dental treatment provided. Contracts with specialist providers (such as hospital or community clinics or specialist practices with specialised dentists undertaking complex procedures or working with complex patients) vary in nature significantly between subregions and between providers. Dental lead commissioners are usually supported by a team of contract managers and administrators as well a local dental network (LDN), a group of clinicians, whose chair is employed by the commissioning team to provide clinical input. In addition commissioners

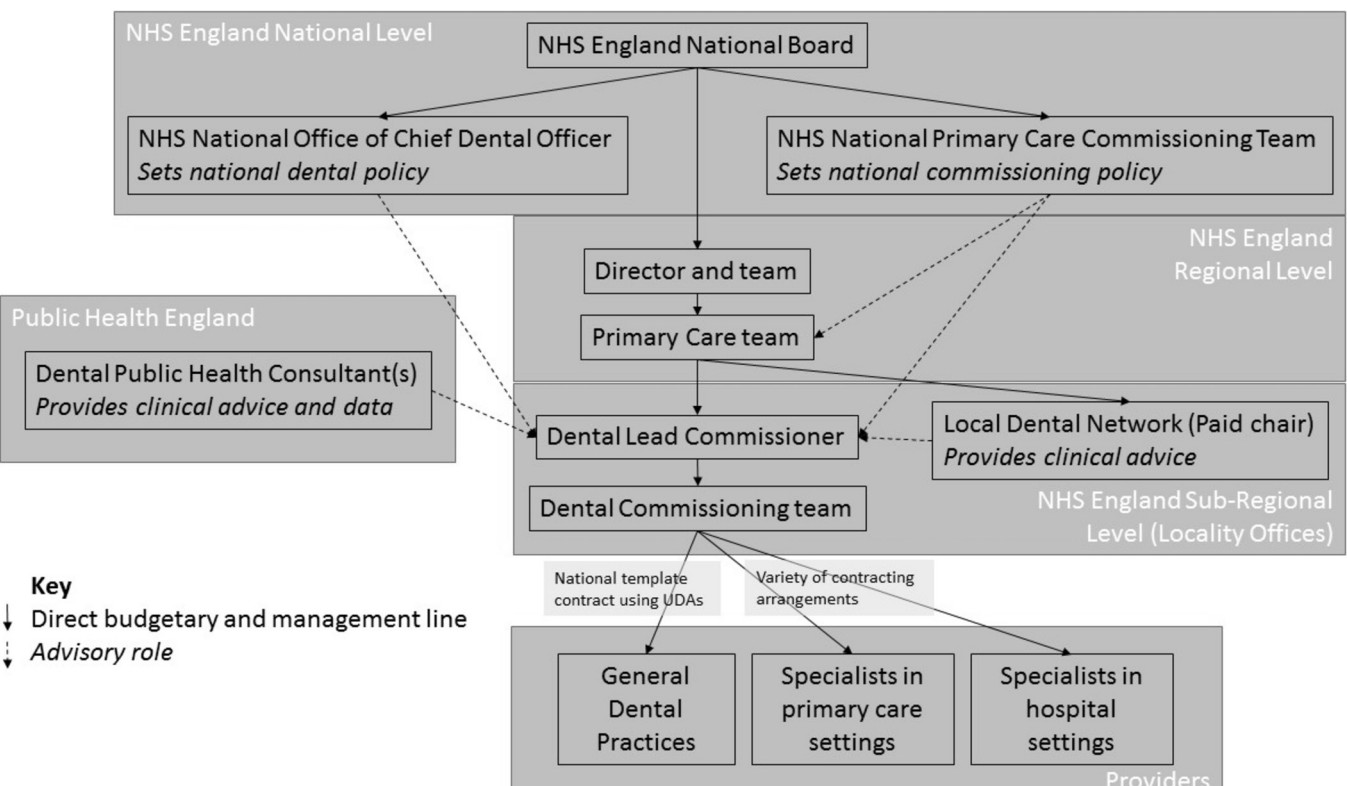

**Figure 1** Dental commissioning structure in the NHS in England. NHS, National Health Service; UDAs, units of dental activity.

rely on advice from consultants in dental public health (CDPH) employed by a separate body (Public Health England), but whose roles are partly to provide advice on dental needs and demand and care provision. The commissioning structure is shown in figure 1.

Given concerns raised in the literature around the use of terms in the area of resource allocation,[16] as part of the context for the study, some key terms will be defined here. The study adopts an economics-based perspective as its theoretical underpinning and was concerned with how priorities are set for resource allocation. In order to allocate resources to be invested in services, in most situations (given the likelihood of a fixed budget size) there must be a release of resources (a disinvestment) from decommissioning other activity. This approach is at odds with some non-economics-based sets of definitions which start from the perspective of disinvestment.[4] The terms commission and investment as well as disinvestment (but not decommissioning) were in common usage within the context and at the time of the study. The terms used in the interviews and this paper therefore are priority setting, resource allocation, commissioning, investment and disinvestment.

This study forms part of an ongoing programme work, the Resource Allocation in NHS Dentistry: Recognition of Societal Preferences (RAINDROP) project, looking to implement economic tools in NHS England dental commissioning, the full protocol of which is reported elsewhere.[17] The larger programme follows a participatory action research (PAR) approach and the study reported here corresponds with the first element of PAR, the 'plan' phase. The methodology employed was semistructured interviews analysed using a framework approach.

### Participants and recruitment

The sample was drawn from the NHS England commissioners responsible for leading dental commissioning in subregional areas (n=31). The study was explained during a commissioning seminar offered to all NHS England commissioners responsible for leading dental commissioning in subregional areas and all those attending were invited to participate. A follow-up email was sent to all dental lead commissioners including those not present at the seminar. One further reminder email was sent to selected commissioners who had not responded at the initial seminar nor to the follow-up email. These were selected purposively with the aim of recruiting further participants to ensure representation of different regions. A written participant information sheet was provided at the seminar and also attached to each email and written consent was taken from those wishing to participate.

### Interviews

Individual semistructured interviews were conducted by one male clinical academic with a background in paediatric dentistry (BDS, MPaedDent) and health economics (PhD) as well as formal training and experience of interviewing for qualitative research (CRV). Participants did not have an existing relationship with the interviewer, but were aware of the interviewer's background and also

the wider context of the research in terms of this being part of a programme about using health economics tools for priority setting in dental commissioning, which was explained at the seminar or at the beginning of the interview for those not present at the seminar. Most interviews (n=10) were conducted in a private room at the commissioners' place of work. However, where it was impossible to arrange this, telephone interviews were used (n=2). Interviews followed a topic guide (available in full as online supplementary file 1), which was initially developed by the wider research team based on previous research and evolved as interviews continued. The topic guide covered experience and role in commissioning, relationships with stakeholders in commissioning dentistry, relationships with dentistry's national team, barriers to commissioning and opportunities and techniques for priority setting in resource allocation. Interviews lasted between 20 and 68 min with a mean of 43 min. The interviews were audio recorded and transcribed verbatim. Additional limited field notes were taken during the interview.

### Analysis
Transcripts were uploaded into NVivo V.11 for management. A coding frame was developed by discussion with the wider team including public representatives. Coding was applied to two transcripts by CRV and GT independently and compared. CRV then coded all other manuscripts and GT reviewed this coding and amended after discussion. A framework analysis[18] was then undertaken, initially by CRV and GT and then with input from the wider team in order to identify emerging themes and concepts.

### Patient involvement
During the development of the study, the National Institute of Health Research, Research Design Service North East Patient and Public Involvement Consumer Panel were actively involved in both defining aspects of the research questions to be addressed and in the development of the project and application.

During the actual conduct of the study, two members of the public sat on the steering group for the wider RAINDROP project and have influenced the ongoing design of the research as well as being involved in the analysis at the design of the coding frame stage and then through discussion of emerging results.

### RESULTS
### Response
The recruitment process resulted in 14 positive responses and 1 refusal to participate with 16 non-responders. It was impossible to arrange an interview with two positive responders and so, 12 interviews were completed. After this initial set of 12 interviews, data were reviewed but as data saturation was noted no further reminders were issued.

In the sample of 12 commissioners interviewed, all regions of England were represented. Further demographic details are not reported due to the small size of the population group and the potential for de-anonymising individuals. In addition, for this reason, only participant number (1–14 as the 2 commissioners who an interview were not arranged for were allocated ID numbers) are given with the quotes below, which are presented to illustrate the findings discussed.

Three major themes arose: methods of priority setting and barriers to explicit approaches; relationships with key stakeholders and advisors; tensions between national and local responsibilities.

### Methods of priority setting and barriers to explicit approaches
Although participants felt there was a real opportunity for undertaking the whole commissioning cycle in dentistry, it was stated that the procuring services and monitoring elements (often referred to by commissioners as 'contracting') of the cycle dominated and that other aspects of commissioning, in particular the strategic planning aspects (which would include priority setting) did not occur in many cases.

> The patch has got bigger, it has got a bit more complex, the resources have got less, so very heavily focussed on what I would call the contracting side, with limited opportunities to do any proper real commissioning. Participant 3

There were several different reasons given for this lack of strategic planning activity. These included a lack of knowledge and skills within the team, a lack of time due to the demands of contracting activity, and having to wait for guidance from the national team. In terms of a lack of knowledge and skills this related both to the specialised knowledge required for dentistry and the lack of commissioning skills (such as understanding resource/budget use, strategic planning, stakeholder engagement, procurement rules and legal requirement as well as understanding of the specific area (dentistry)) in teams. There was some evidence that participants felt that teams had detailed dental knowledge but less generic commissioning skills.

> We have had external consultants working for us which is fine but … thinking about the fact that we have got a team of 30, it's thinking should we really be trying to upskill the team? And, yet we are in a certain place and we have had to buy in some subject matter expertise. Participant 6

The lack of knowledge and time was magnified in some areas where teams covered small areas but were designated as generic 'primary care' teams covering multiple types of providers (general medical practitioners, pharmacists, optometrists as well as dentists), with individuals in the teams dealing with multiple provider types and the 'dental' lead actually being a 'primary care' lead. In other areas teams, individual team members and the lead were allocated specifically to dentistry but across a larger area. Although it appeared that dental specific, large area

teams engaged more in the strategic planning activities, those in more generic but smaller area teams felt they could develop better relationships with their providers.

> My personal view is I think you need specialist teams, particularly for dentistry, particularly because you don't have the support … Historically in [pre 2012 structure], I could have accessed health and safety, infection prevention, a whole raft of clinical governance people. [I] can't do that now because if it sits in [different area of structure] and they don't want to know about dental. So I do think you need a specialist team because we will have an awareness of those things, as a minimum. People are learning all the time but actually, you don't get below the surface. Participant 8

Whether the team was dental specific or not, teams were being asked to cover increasingly large geographical areas as teams merged and this was exacerbating the lack of time and resources. In addition, there was an increasing burden of contracting activity in terms of monitoring of current contractors and legal challenges.

When questioned directly about priority setting aspects of the strategic planning, participants had clear ideas about some of the specific areas with resource allocation problems.

> We have got [good] guidelines for orthodontics but then you still hear comments about, "Well, should we really be prioritising all of that when we have got all these children with decay and we have got all these elderly people in nursing homes with complex restorations that are just falling apart now because they are old and they are struggling with health and other things that impact then on the roll-out?' Participant 2

However, when asked how they set priorities for investments and disinvestments in different services, no commissioners mentioned undertaking any proactive priority setting and no systematic attempts were mentioned. Rather, they felt that their resource allocation, where it did happen was reactive to oral health needs assessments, pressure from clinicians and pressure from politicians.

> We get representation. We get MPs 'we want a dental practice here, we want that here', the consultants saying…, this consultant, that consultant, orthodontists all saying 'it should go to orthodontics'. How we're going to manage it; I think the group's struggling with that. Participant 13

The advice or pressure from clinicians was described as conflicting and potentially self-interested.

> Well we have consultants turning up who say that restorative dentistry is the priority and then we have Public Health and our LDCs' representatives saying that … dentists on the high street, need support. … and we have community teams turning up wanting lots of extra money in their contracts. So they're all there looking after their own areas. Participant 1

There was a recognition that expertise and tools to aid priority setting would be useful, especially in the light of limited skills in commissioning teams, but also that national direction on what the priorities for investing and disinvesting should be would be useful.

> Yes, so, …because we can't do everything, we only have a limited budget, then a clearer steer on some of these issues would be helpful but is it a national policy that we continue to do this or not? Participant 2

Finally, the national contract used between commissioners and primary care dental providers, based on the generation of UDAs and in general held in perpetuity by providers, was seen as a barrier to reallocating resources (although some resource may be released if providers fail to meet their contractual obligations, request a renegotiation or terminate a contract and secondary care contracts are generally time limited).

> There's very little commissioning that you can do at the moment, largely because the large bulk of our contracts are in non-time limited [general primary care dentistry] contracts. Participant 4

However, one commissioner did report that the contract and UDAs had not held them back, and they had found ways around this, such as using other minor, little used clauses in the contract.

> You can't use a UDA [for allocating resources for priorities]. For me it's like a bean counting system, but we have been able to use the contracts innovatively and use some of the public health element within the contracts. Participant 9

### Relationships with key stakeholders and advisors

The commissioners articulated the need for good clinical advice to support their commissioning and priority setting activities.

> [Without the support of PHE and clinicians] we would just be a bit of an island and not be successful, I don't think, in what we do. But I think it's that teamwork and that clinical engagement that helps. Participant 9

It seemed that the advice came from three different sources, which were different for different commissioners. First, some relied on their LDN and its chair. However, there was a feeling that there was still a lack of clarity about the nature and power of the LDN and its relationship to the commissioner.

> I think it's still bedding down, you know, the [LDN], you feel like it's an organisation without any teeth, it's very much more a talking shop. I think the conflict of interest makes things extremely difficult. Participant 11

> It's one of those things that's a little bit frustrating, from my perspective; when we had the pilot, [LDN], it very much sat within my control, and now [that it's

not a pilot but established,] I no longer have control and, being a control freak, that's a little bit difficult to cope with … we're also trying to work out how we work together, what they do, what we do… we're trying to get that understanding of what the LPN is actually there to deliver. Participant 14

Second, many commissioners looked to their CDPH colleagues for clinical advice, but the strength of relationship depended on the availability in the subregional area and also whether the DPH team were colocated with the commissioning team.

Also, we know [the CDPHs] are being spread a lot more thinly, they're being brought in to do more national bits of work, which is great, potentially, for the national bit but, for us, it's beginning to feel like a bit of a hole. I think we feel it more because we've been very active, in terms of working with them and using them; I think other areas haven't got that history. I know in [neighbouring sub-regional area], they're thinking, 'Great, we've got some support now, which we never had before.' Participant 13

Finally, some commissioners relied on dentists employed directly by the commissioning team as dental practice advisors (DPAs). These individuals are trained and employed to advise on contract monitoring issues and provide advice on quality issues with single providers, but some commissioners were using them in alternative ways for strategic planning.

We have very good clinical advice from [the DPAs] and all sorts of advice really. Obviously not just clinical. Participant 7

It appeared that those with good relationships with LDNs and CDPHs felt more able to undertake priority setting, resource allocation and the whole commissioning cycle in general.

I'm really enthusiastic about dentistry and the work that we do here in [the region] and I don't think we could do it without the support of [CDPHs], without the support of our [LDN]. Participant 9

There was a general frustration with the limited relationship (either in terms of strength of relationship, quality of relationship or absence altogether) with several other stakeholders namely, patients, Health Education England (with a responsibility for ongoing education of the dental team and workforce planning) and non-dental commissioners (who may have several overlapping areas such as emergency and out of hours care, care of special needs groups and health promotion).

We really did struggle, we wanted actual patient reps on both of the [LDNs] but we really struggled with that. . so that is going to be a challenge for us, to identify the right patients that don't have a vested interest. Participant 4

## National versus local responsibilities

Participants noted both advantages and disadvantages to having a nationally commissioned system but also stated that the reality was confusing with a lack of clarity over whether decisions (not only relating to resource allocation but more widely) were actually taken at national and local level in reality.

If the view is that NHS England is going to do it once nationally, then that's what we need to do; we need to stop faffing about with local this and local that, and actually make it a national organisation. If they're looking to devolve it, then we need to think a bit differently about how you can devolve those decisions within a national framework, and I think you would need very clear expectations about how that would work, and what rules you were working within. Either way, there needs to be more direction. Participant 14

We get very mixed messages about, well, are we a local office working within NHS England and national structure, or are we the local accountable body for implementing the NHS England policy, that has got nothing to do with the centre, and they fluctuate between those views, really. Participant 13

Positive aspects of nationally led commissioning included both the need to only do things once rather than repeat a process in each subregion and also the consistency that came with having one way of doing things.

We still have 31 different ways of doing things in some ways unfortunately and it's great to be able to get a piece of paper in a meeting, or say, 'Just look at the link, that's what we are using that underpins everything now.' Participant 6

However, the downsides were that there was a limit in local flexibility and that local commissioners had to wait for central decisions, sometimes beyond when a decision needed to made locally. There was real concern that it was not clear where the responsibility or power for decisions rested and so some decisions were not taken at all.

We've got all of our community dental services [contracts] that are going to be expiring [soon after interview]. Because we haven't got the [national] paediatric [dentistry commissioning] guide yet, we don't want to go out to commission new services until we've got the paediatric guide because we could commission something that isn't in line with that. Participant 4

For the dental lead commissioners in subregional teams, there was often a confusion between who they were answering to, whether this was the regional director and board (who held the budget) or the national NHS England dental team, who were sometimes giving different messages. There was also a recognition that the national team was too small to be effective.

We've got this difficulty in terms of local and national because, actually, a lot of what we do within dentistry, we will go directly to the national team and then, locally, the … Directors are saying, 'Well, hold on a minute, we don't know anything about this.' Participant 14

[In the national team] With three people doing [dentistry], and they don't just do dentistry either, they need additional support because [with] the structure that [was] set up, [they] either give us the autonomy to do things locally, in which case let us have the resources to do that. Or, if you want more of a central steer, give them the resources so that they can do it as well. Participant 4

## DISCUSSION

This exploratory qualitative study using semistructured interviews with commissioners found that NHS England dental commissioners felt that there were problems with current resource allocation and that proactive, systematic priority setting for the allocations did not happen. Respondents felt that barriers to priority setting activity included a lack of time, a lack of knowledge, a lack of skills and inflexible contracting arrangements. Commissioners relied heavily on good links with LDNs and CDPHs for clinical input into commissioning decisions but these links varied. Where links were strong, commissioners felt more able to undertake priority setting. When it did occur, the priority setting process was driven by political and clinician pressure as well as needs assessments, rather than more explicit processes. As described in the introduction, we would see these explicit processes as 'making decisions whereby options for change (both investments and disinvestments) are assessed against each other on the basis of defined criteria, weighted for relative importance, and considered also against the cost of each such option.'

There was also a tension between the national team and the subregional commissioners. Dentistry is, in theory, a nationally commissioned service yet in reality budgets and therefore resource allocation decisions are devolved to regional level, generating confusion about where authority and responsibility lies.

As described in the introduction, the study is unusual in conducting in-depth interviews with those actually making resource allocation decisions (ie, subregional commissioners) rather than at a higher managerial level (eg, board members, financial directors or national leaders, who may also need to undertake priority setting and resource allocation decisions but at a more strategic level) and also looked at decision-makers across a whole country rather than in one area or a small number of case study areas. In addition, by adopting qualitative methodology, the reasons for adopting specific approaches to resource allocation could be explored in depth. However, due to the limited total number of individuals involved in dental commissioning in England, the numbers were small (even though 40% of this population were surveyed), although it appeared data saturation was reached in the interviews. Additionally, one specific area of healthcare commissioning (dental) was studied with an unusual configuration compared with other areas of health in the country studied, which may lead to specific findings not relatable to other areas of healthcare commissioning. Nonetheless, a considerable amount of NHS national spend relates to dentistry and so this is a worthwhile area to study.

Compared with the two of the studies noted in the Introduction section,[3] [11] where there was a move towards some use of priority setting tools, the findings here indicate that there was no use within the setting of this study, which does fit with what was found in Australian study described in the Introduction section.[13] This may reflect the fact that the specific health area has not yet engaged with these tools or that these tools are only used at the more strategic/managerial levels covered in the two other studies. It is interesting to note that the most important resource used for priority setting here, needs assessment, was also the most frequently used in the questionnaire-based study of Robinson et al.[3] Looking at the barriers to priority setting, this study had strikingly similar results to a similar study undertaken with dental commissioners in one English region 10 years previously,[19] with consistent themes of contracting activity dominating commissioning, frustration with a lack of national direction and limited-resource reallocation possibilities due to the nature of contracts. At the time of the previous study, dentistry was commissioned within much smaller local organisations (primary care trusts) with full budgetary and decision-making control and local structures providing all advice and support in house locally. However, there was still a national dental lead (the chief dental officer) but this role was only advisory to the independent local organisations. The similarities in findings suggest that despite markedly different structures being in place now, the challenges remain the same and have not been addressed in over a decade. The findings in this paper also fit with the Sustainability in Health care by Allocating Resources Effectively programme in Australia in a public hospital and community care group, where a disconnect between senior managers and those at more local decision-making levels was observed and barriers of time and expertise were noted.[13] [20] This suggests that the findings may be generalisable beyond the English and dental settings.

The case for explicit prioritisation in resource allocation decisions is made in the introduction but does not appear to be occurring in the context of this study. Given the reported lack of skills in and time for priority setting, frameworks such as PBMA, MCDA, A4R and STAR may be useful in simplifying the process and making explicit priority setting more accessible. The study also suggests that good links with clinical advice sources should be prioritised and grown, given that commissioners felt more able to set priorities when good links were in place and there may be a case for integrating this clinical advice fully into commissioning teams. In addition, the findings

suggest that where a national system is operationalised at subregional levels, care needs to be taken to be explicit about where power and responsibility lie.

In terms of future research, there is a need to consider how more explicit priority setting mechanisms including the use of tools can be embedded into day-to-day operational commissioning decisions, which may be through better illustrating their utility as well as designing appropriate training and simplifying their use and adoptability.

## CONCLUSION

Commissioners recognised a need for resource reallocation but relied on clinical advice and needs assessment in order to set priorities. Links with clinical advice sources were, therefore, seen as key. A lack of time and expertise were barriers to more formalised priority setting and resource reallocation, as was the structure of the commissioning system, with tensions between local and national control and the contract for the service in question, dentistry, which perpetuated historical allocations.

**Author affiliations**
[1]Centre for Oral Health Research, Newcastle University, Newcastle upon Tyne, UK
[2]Child Dental Health, Newcastle Upon Tyne Hospitals NHS Foundation Trust, Newcastle Upon Tyne, UK
[3]Social Business and Health, Glasgow Caledonian University, Glasgow, UK
[4]Department of Nursing, Midwifery and Health, Northumbria University, Newcastle upon Tyne, UK
[5]Faculty of Health and Life Sciences, Northumbria University, Newcastle upon Tyne, UK

**Acknowledgements** The authors wish to acknowledge the large contribution of the late Jimmy Steele CBE to the genesis of the idea of this study as well as a major contribution to the design and set-up. In addition, the authors wish to thank the Oral and Dental Patient and Public Involvement Group at Newcastle University and particularly Irene Soulsby and Ian Armstrong for their advice on the study design, input into analysis and their ongoing support of the wider study. Finally, the authors wish to thank NHS England, the Office of the Chief Dental Officer (England) and the individual commissioners for access and giving up their time for the interviews.

**Contributors** CRV was the lead for the conception and design of the study, collected the data, was involved in the analysis and drafted the manuscript. GT was involved in the analysis and critically revised the manuscript. CD was involved in the conception and design of the study, was involved in the analysis and critically revised the manuscript. JG was involved in the analysis and critically revised the manuscript. RH was involved in the conception and design of the study, was involved in the analysis and critically revised the manuscript. KC was involved in the analysis and critically revised the manuscript. CE was involved in the conception and design of the study, was involved in the analysis and critically revised the manuscript. All authors approved the final manuscript.

**Funding** The study is funded through a Clinician Scientist Fellowship for CRV from the National Institute for Health Research.

**Disclaimer** The views expressed are those of the author(s) and not necessarily those of the NHS, the NIHR or the Department of Health.

**Competing interests** None declared.

**Patient consent for publication** Not required.

**Ethics approval** Ethical approval has been granted for the study by Newcastle University Ethics Committee (Ref No. 00873/2015). In addition, approval was gained from the Health Research Authority to undertake research within the NHS (Ref No. IRAS 184704).

**Provenance and peer review** Not commissioned; externally peer reviewed.

**Data sharing statement** No additional data are available.

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
