## [Reviewer comments · BMJ Open]

ARTICLE DETAILS

TITLE (PROVISIONAL)	How does priority setting for resource allocation happen in commissioning dental services in a nationally-led, regionally-delivered system? A qualitative study using semi-structured interviews with NHS England dental commissioners.
AUTHORS	Vernazza, Christopher; Taylor, Greig; Donaldson, Cam; Gray, Joanne; Holmes, Richard; Carr, Katherine; Exley, Catherine

VERSION 1 - REVIEW

REVIEWER	Claire Harris Monash University Australia
REVIEW RETURNED	09-Jul-2018

GENERAL COMMENTS	How does priority setting for resource allocation happen in commissioning healthcare services in a nationally-led, regionally-delivered system? A qualitative study using semi-structured interviews with commissioners. Thank you for the opportunity to review this paper. Priority setting for resource allocation is an important topic and this investigation adds to the understanding of this process in healthcare decision-making overall and in the field of dentistry in particular. The paper is generally well-written, but I feel more detail of the NHS processes is required to enable understanding for those outside this system. TITLE I suggest that a reference to dentistry is added to the title for clarity. Perhaps replace 'healthcare services' with 'dental services' to remove any ambiguity? TERMINOLOGY I think it would be really helpful for the reader, and would strengthen the authors' arguments, if the terminology was well defined and then used consistently throughout the paper. The title and several other references in the manuscript describe the topic as 'priority setting for resource allocation' which suggests that 'priority setting' and 'resource allocation' are different things and that priority setting is part of the process of resource allocation, which I agree with. However these terms also seem to be used interchangeably, for example P8 L53-55, P9 L17. The term 'prioritisation' seems to be used synonymously with 'economic approaches to priority setting' throughout the paper. From my understanding of the resource allocation literature, I think that these are two different things: 1) generic processes for prioritisation and decision-making which may or may not include economic criteria and can be used by all decision-makers and 2) 'economic approaches to priority setting' which are specific
--

methods based on health economic principles which require the expertise of health economists in their application.

In the Introduction (P3 L30-35) the authors state 'Specific economics-based priority setting tools have been developed, such as MCDA, A4R, PBMA or STAR.' My understanding is that PBMA is one of several 'economic approaches to priority setting' and MCDA, A4R and STAR are examples of generic decision-making processes. For example, none of the 4 components of A4R relate to economics or priority setting and are frequently referred to as 'ethical factors' in decisionmaking. While all of these methods are relevant to the background to this study, the authors may wish to refine the wording to reflect current terminology. A recent literature review on this topic may be useful [1]. The review also lists a number of other methods and tools for both economic approaches to priority setting and generic prioritisation/decisionmaking (including software), plus references for decision-making criteria used in resource allocation in a range of contexts.

The term 'reallocation' (also written as re-allocation) is mentioned in the Introduction and again in the Discussion but not mentioned in the Title, Methods or Results. Is the paper about resource allocation and reallocation, were participants asked about reallocation specifically, and if so, what did they say?

Disinvestment is also mentioned in the Introduction but not in the rest of the paper (were participants asked about this?).

Since this is a paper about commissioning, I wonder why the term 'disinvestment' was used rather than 'decommissioning'. There is considerable debate in the resource allocation literature about definitions for these terms; sometimes one is used to define the other, sometimes the definitions overlap, sometimes they are mutually exclusive [2]. Although there are a range of definitions, it could be said that 'decommissioning' tends to be used more for funding decisions related to health services and 'disinvestment' used in decisions about monetary and non-monetary resources for removal, reduction or restriction of individual clinical practices within health services. Based on these definitions, decommissioning and disinvestment will have quite different issues and specific barriers. It would be helpful to know what is being referred to in this paper.

I suggest that appropriate terms are chosen to convey the desired concepts, that the terms selected are defined, and then used consistently within the manuscript.

ABSTRACT

Three themes are noted in Results. The first is written with a capital letter. Given the long sentence with lots of punctuation, I think it would help the reader if the first words of the other two themes were also capitalised or a numbering system was introduced (eg 1) methods....., 2) relationships.....etc), or both. While I think the current wording is technically correct, I had to read it twice to pick out the three themes.

INTRODUCTION

For their overview of use of explicit mechanisms for priority setting the authors may wish to know about several other studies that have reported use of decision-making tools for resource allocation – the literature review noted above includes eight (four of which were UK based) [1]. The Australian SHARE Program investigated resource allocation decisions at all levels within a regional health service (ie not just board level) [3, 4] and investigated and applied explicit prioritysetting mechanisms [5]. Many of their outcomes are consistent with those reported in this manuscript which would help to strengthen the potential generalisability of the findings.

METHODS

Context

The context is well described however, as someone unfamiliar with the UK system, I found it hard to follow some of the details when they were discussed later in the paper. This is a minor point, but it would have helped me to be able to refer to a diagram of the hierarchy for commissioning dental services.

While I understand that those in the field may be confused about where decisions are made (national or local), and this finding is also consistent with the SHARE program findings [3], I wonder if there is documentation about who should be responsible for what. Are the local decision-makers unaware of what is done at national level and what is required of them at local level, or is there no guidance on this at all? Does the shortcoming exist at national level (absence of guidance) or local level (lack of awareness of guidance) or is it something else?

I think the acronym RAINDROP should be written in full.

Recruitment

'...delivered to all NHS England commissioners...' '...including those not present at the seminar'

Another minor point, but if some were not present then the seminar was not delivered to ALL the commissioners. Perhaps it was 'offered to all'?

Sample

There is considerable duplication between Recruitment and Sample. Perhaps combine the subheadings? Or have Sample first to describe all dental commissioners and then Recruitment to note the seminar, personal and email invitations, and follow up?

I think it is incorrect to say 'sample' when all members of the relevant group are invited ie there was no sampling from the target group. Perhaps 'participants', 'interviewees' or 'target population'? I think the second half of this paragraph regarding response rate should be in Results.

Interviews

Were interviewees specifically asked about reallocation and disinvestment as noted above?

'...were aware of the interviewer's background.....wider context....'

Does this mean the interviewees were made aware of the interviewer's background and wider context at the start of the interview or that they had previous knowledge of this?

I think a more detailed outline of the interview schedule/proforma should be provided. The list of topics does not cover the level of detail about the questions that were asked or prompts used. This raised questions for me as I read through the Results. This could be provided in a supplementary file if the word limit precludes inclusion in the manuscript (although adding it in a table could also get around this).

I am interested in the reasoning for separating male and female respondents and identifying responses by gender. Identifying subgroups of health service staff is not unusual, but it is usually done because of the potential for different groups to respond differently based on their roles; for example doctors/nurses, clinicians/managers, senior/junior, hospital/community. It may be important to separate responses from male and female patients in clinical trials because of gender specific factors that influence health, but I am not aware of such factors in health service employment when all participants are in the same role. Was there a hypothesis that male and female commissioners may respond differently? If so, I think it is important that this is documented in the Methods and Results and then considered in the Discussion.

What is the evidence to support the hypothesis? How do the results compare to what is known on this topic?

Analysis

After emerging themes were defined ‘...results were discussed with a sub-sample of participants...’ To what purpose?

Were specific aims for the discussion presented to the sub-sample? What were the aims? Was this process structured? How did this process influence the results?

RESULTS

I think there should be a subheading for Response. This should include the response/participation rates and details regarding gender if this is considered to be a factor that would influence the responses. How many commissioners were men/women? Does the response rate reflect this ratio? Was there any systematic difference in their responses? What was it? What might that mean? There are 17 quotes, only 5 are male and 3 of these are from the same person. Is this representative of the participation ratio?

Methods of priority setting

As I am unfamiliar with the NHS models, it would have been helpful to have a figure of the commissioning cycle. In my experience ‘methods of priority setting’ and ‘strategic planning’ are not the same thing and I would not put the issue of being unable to undertake the whole commissioning cycle or how commissioners undertake strategic planning under this subheading. However my confusion may be due to lack of familiarity with the processes in the commissioning cycle. I suggest further elaboration on the cycle, and how strategic planning is a method of priority setting in this context, for readers outside the UK.

P8 L23-28. I am confused as to why dental commissioners are covering other providers, could this be explained please. I found the wording ‘...although others saw this as an advantage as they generally had smaller geographic areas...’ to be ambiguous. Does ‘this’ refer to being asked to cover multiple providers ie the advantage is the coverage of multidisciplinary activity? Or is the advantage having smaller geographic areas? Or developing better relationships with providers? From my reading of the current wording, it could be possible that having a smaller geographic area and better relationships with providers are both advantages but that commissioning for multiple providers is still a disadvantage – so it’s not one or the other.

P8 L46. ‘...the increasing geographic area that teams were being asked to cover...’ This seems to be inconsistent with the paragraph above about smaller areas. Perhaps additional elaboration would make this clearer.

P9 L3. Just checking....should ‘nice’ be ‘NICE’?

P9 L16-18. ‘...when asked how they would prioritise the areas, no commissioners mentioned either specific systems or tools...’ What are ‘the areas’? I assume this is in response to an open-ended question. Were there any follow up questions asking commissioners if they had used any methods or tools, or any questions that asked them about specific tools? If so, I think this needs to be added. If not, how can the authors be sure that specific methods just didn’t come to mind or that the commissioners didn’t consider the method they used to be a ‘system or tool’? Why is the question phrased to identify future activity (how WOULD they prioritise) rather than reflect past activity (how DID they prioritise)?

P9 L44-51. The reference to 'national direction' in the text suggests that national direction to use expertise and tools in priority setting would be useful. But the reference to 'national policy' in the quote is about 'continuing' to do something. If, as previously stated, the commissioners have not been using specific methods and tools then this reference can't be about national policy to continue to use methods and tools they haven't been using. P9 L48. Just checking...is there a typo or missing words in this quote before or after 'then'?

P10 L4. If the national contract is held in perpetuity by providers, what local commissioning can occur? P10 L17. The primary care contract is based around UDAs (P4 L35-36) but this quote states 'You can't use a UDA.' These points are confusing, I think further elaboration is required to explain what seems to be inconsistencies.

Relationships with key stakeholders and advisors

P10 L39. 'It seems that this came from three different sources...' I wonder if 'this' should be 'the advice'. The topic of the previous paragraph is the 'need for advice', hence 'this' in the following paragraph would be the need for rather than the actual advice. From my reading of this paragraph I suspect it is the latter. The findings are reported as 'confusion and disagreement' but only one quote is given. The current quote does not convey confusion and at least two are required to demonstrate disagreement.

P11 L5. What does 'level of relationship' mean? Should 'depending' be 'depended'?

P11 L23-26. Why do commissioning teams employ dental advisors for a purpose other than commissioning? And why is this a problem for the commissioning process? There is no quote for this finding.

P11 L30-31. How do good relationships with LDNs and CDPHs create better opportunities for priority setting, resource allocation and commissioning in general? I'm not sure what 'opportunities' means in this sentence or how opportunities are affected by these relationships. There is no quote for this finding.

P11 L36-39. What does 'limited' mean here – absent, insufficient, poor quality, something else? Why are non-dental commissioners stakeholders? National versus local responsibility

This is referred to as 'Tensions between national and local responsibility' in the abstract. Should it be 'responsibilities'? I think the theme should be written the same way in both places.

As noted earlier, I think more elaboration about the process is required. Is there an established national process that local commissioners are unaware of, or is it ambiguous, or confusing, or absent, or something else? There is a suggestion in the quote that it might be 'devolving' – is it? Is this the cause of the confusion? It's hard to tell what are facts that could be substantiated (eg that there is/isn't clear documentation on the responsibilities at national/local level, or that the national process is/isn't to be devolved to local bodies) and what are the perceptions of local commissioners (which could be right or wrong).

P12 L42-48. This seems to be a good quote to illustrate national/local confusion but not to illustrate limited flexibility or delays in central decisions.

P12 L53-55 and P13 L8. Are the 'dental leads' dental commissioners? Are the 'local teams' the 'dental care teams'? What is the 'hierarchy locally'? What is the relationship between dental commissioners and dental care teams? Who are the 'Directors'? A figure with these relationships would be really helpful to those unfamiliar with NHS structures.

P12 L15-21. Just checking....is there a typo or missing words/punctuation in this quote? The words '...the structure we were set up either give us the autonomy....' don't make sense to me. DISCUSSION

Several claims are made that I could not find in the Results section.

- 'The ...study ...found ...commissioners felt that resource re-allocation was needed but did not happen often or systematically.'
- 'Where links were strong, more priority setting seemed to occur.'
- '...tension between the national team and the regional commissioners.' (There is no mention of the national 'team' only responsibilities)
- '...frustration with national direction not aligning with local needs...'
- '...good links with clinical advice should be prioritised and grown...'

P13 L51-53. '...with those actually making resource allocation decision rather than at a higher managerial level...' The interviewees were dental commissioners. What would be the higher managerial level of people involved in commissioning dental services? Why would they be interviewed rather than the dental commissioners?

P14 L3. The wording '...reasons underlying resource allocation could be explored in depth' is ambiguous. Reasons for what? Is this referring to resource allocation decisions, the resource allocation process, or something else? No criteria used in resource allocation decisions are mentioned. Some barriers to use of specific tools for prioritisation are mentioned, but the process of resource allocation [3], or any reasons related to how resource allocation is practised are also not mentioned.

P14 L10-13. While there are still some limitations to the generalisability of this study, some general findings (lack of time, skills, etc) are found in many other studies [1] and more specific findings (confusion about who makes final decisions, lack of use of prioritisation tools) were also found in the SHARE studies [3, 5] and others [1], which strengthens the findings herein. If it was established that national direction does not align with local needs, this has also been reported by others

[6]. P14 L35. '..markedly different structures...' Different to what? How are they different? How would you expect the differences to impact on the findings?

P14 L 43. Why are PBMA, MCDA, A4R and STAR recommended when Public Health England has developed a specific tool for this purpose?

CONCLUSION

P15 L12. '[Lack of] time and expertise were barriers...'

I have made several references to papers from the 'SHARE' Program in which I was involved. I am not suggesting that the authors need to include these as references, but have cited them because I know they are a source of information for the points I have made. There may be other suitable sources.

1. Harris C, Green S, Elshaug AG. Sustainability in Health care by Allocating Resources Effectively (SHARE) 10:

Operationalising disinvestment in an evidence-based framework for resource allocation BMC health services research. 2017. doi:<https://doi.org/10.1186/s12913-017-2506-7>.

2. Harris C, Green S, Ramsey W, Allen K, King R. Sustainability in Health care by Allocating Resources Effectively (SHARE) 9:

	Conceptualising disinvestment in the local healthcare setting BMC health services research. 2017. doi:https://doi.org/10.1186/s12913-017-2507-6. 3. Harris C, Allen K, Waller C, Brooke V. Sustainability in Health care by Allocating Resources Effectively (SHARE) 3: Examining how resource allocation decisions are made, implemented and evaluated in a local healthcare setting BMC health services research. 2017. doi:https://doi.org/10.1186/s12913-017-2207-2. 4. Harris C, Ko H, Waller C, Sloss P, Williams P. Sustainability in Health care by Allocating Resources Effectively (SHARE) 4: Exploring opportunities and methods for consumer engagement in resource allocation in a local healthcare setting BMC health services research. 2017. doi:https://doi.org/10.1186/s12913-017-2212-5. 5. Harris C, Allen K, Brooke V, Dyer T, Waller C, King R et al. Sustainability in Health care by Allocating Resources Effectively (SHARE) 6: Investigating methods to identify, prioritise, implement and evaluate disinvestment projects in a local healthcare setting. BMC health services research. 2017. doi:https://doi.org/10.1186/s12913-017-2269-1. 6. Harris C, Garrubba M, Allen K, King R, Kelly C, Thiagarajan M et al. Development, implementation and evaluation of an evidence-based program for introduction of new health technologies and clinical practices in a local healthcare setting. BMC health services research. 2015;15(1):575. doi:10.1186/s12913-015-1178-4.
--	---

REVIEWER	Martina Garau Office of Health Economics
REVIEW RETURNED	08-Aug-2018

GENERAL COMMENTS	Thanks for the opportunity to review this article which explores and provide evidence on the important topic of lack of systematic and efficient priority setting for the provision of part of NHS services. The focus on dentistry is well justified and represent a useful example of disconnect between national and local decision making and call for a need for more explicit resource allocation processes. The method is well explained and appropriate for the research question. The discussion provides a good comparison of the results with previous literature. The article, however, introduces a number of concepts and terminology (probably used by some of the interviewees) that needs further explanation and analysis. Given the variety of expertise and nationality of BMJ readers, it would be useful to clarify what is meant by “explicit” priority setting processes and tools, and to explain what the role of NICE guidelines is in decision making, for example. The themes identified are relevant and they provide valuable evidence of the existing issues. However, more analysis or interpretation of interviewees’ quotes would improve the quality of the article. Finally, the article provides some clear messages but many sentences are not easy to follow and understand. Therefore I suggest re-drafting part of the text to increase clarity and readability. Some of the sentences are pointed out in my specific comments below. Specific comments:
---

	The title should specify that the study is focused on dentistry services in England The text in the introduction is cryptic in parts and might benefit from some more specific examples and recent references in the literature, for example: Page 2, line 26: sentence might need re-drafting as not very clear (“where the variable but key nature of relationship with clinical advice sources was revealed”) Page 3 line 19: the sentence with “prioritisation was undertaken” should specify the setting and the timing the authors refers to Page 3 line 26: it might be useful to specify what “traditional economic evaluation” techniques are (with a more recent/better known paper referenced) and maybe refer to their use within the NICE programmes, given that the audience might include non-economists Page 3 line 50: Karlsberg Schaffer et al. also published a study on decision making in Scottish NHS boards (Karlsberg Schaffer et al. (2015) Local health care expenditure plans and their opportunity costs, Health policy Volume 119, Issue 9, Pages 1237-1244). More generally, unless a literature review was conducted, it might be better to soften the tone of the sentence in line 50 (“two notable exceptions”). Page 4 line 15: it is not totally clear the difference between board and decision making level. In the Karlsberg Schaffer et al. articles board is actually meant to represent the decision making level so need further explanation. Page 4 line 32: “is” should be “are” Page 4 line 28: is it possible to provide an example of “sub-region?” Given that the title allude to a possible tension between national and local decision making, the issue should be better presented in either the introduction or the context. Page 5 lines 17 and 34: here the word “regional areas” is used while on page 4 line 28 the word “sub-regional” is used Page 7 line 24: is there a reason to indicate gender in the quotes? Page 7 line 43: what is the “contracting element”? and “commissioning skills”? If these concepts/wording were used by interviewees, there is a need to explain what they are to help the readers Page 10 line 10: what does “GDS contract” stand for? Page 10: consistency is needed in the use of Local Dental Network, LDT, and LPN Page 13 line 42: the sentence in bracket explaining what explicit processes are is key so should not be in bracket and further expanded Page 13 line 31: through the text the authors use the words “allocating” and “re-allocating” but it is unclear if there is a difference between the terms
--	--

VERSION 1 – AUTHOR RESPONSE

Reviewer(s)' Comments to Author:

Reviewer: 1

Reviewer Name: Claire Harris

Thank you for the opportunity to review this paper. Priority setting for resource allocation is an important topic and this investigation adds to the understanding of this process in healthcare decision-making overall and in the field of dentistry in particular. The paper is generally well-written, but I feel more detail of the NHS processes is required to enable understanding for those outside this system.

TITLE

I suggest that a reference to dentistry is added to the title for clarity. Perhaps replace 'healthcare services' with 'dental services' to remove any ambiguity?

The amendment suggested has been made to the title in addition to clarification of the setting.

TERMINOLOGY

I think it would be really helpful for the reader, and would strengthen the authors' arguments, if the terminology was well defined and then used consistently throughout the paper.

The title and several other references in the manuscript describe the topic as 'priority setting for resource allocation' which suggests that 'priority setting' and 'resource allocation' are different things and that priority setting is part of the process of resource allocation, which I agree with. However these terms also seem to be used interchangeably, for example P8 L53-55, P9 L17.

The term 'prioritisation' seems to be used synonymously with 'economic approaches to priority setting' throughout the paper. From my understanding of the resource allocation literature, I think that these are two different things: 1) generic processes for prioritisation and decision-making which may or may not include economic criteria and can be used by all decision-makers and 2) 'economic approaches to priority setting' which are specific methods based on health economic principles which require the expertise of health economists in their application.

The authors agree with the definitions proposed by the reviewer and the introduction has been amended to clarify these. In addition, the terminology has been defined in the Context section and these different approaches have been more carefully referred to throughout the paper.

In the Introduction (P3 L30-35) the authors state 'Specific economics-based priority setting tools have been developed, such as MCDA, A4R, PBMA or STAR.' My understanding is that PBMA is one of several 'economic approaches to priority setting' and MCDA, A4R and STAR are examples of generic decision-making processes. For example, none of the 4 components of A4R relate to economics or priority setting and are frequently referred to as 'ethical factors' in decision-making. While all of these methods are relevant to the background to this study, the authors may wish to refine the wording to reflect current terminology. A recent literature review on this topic may be useful [1]. The review also lists a number of other methods and tools for both economic approaches to priority setting and generic prioritisation/decision-making (including software), plus references for decision-making criteria used in resource allocation in a range of contexts.

The authors agree that some of the examples given are not economics-based and so the text has been amended to clarify this. The addition of the SHARE literature review is a useful suggestion and this is now included alongside an acknowledgment of the wider range of frameworks and tools.

The term 'reallocation' (also written as re-allocation) is mentioned in the Introduction and again in the Discussion but not mentioned in the Title, Methods or Results. Is the paper about resource allocation and reallocation, were participants asked about reallocation specifically, and if so, what did they say?

The terms reallocation and allocation were used synonymously throughout the paper as, within the fixed budget envelope, any allocation was as a result of reallocation. To simplify reading the term allocation has now been used exclusively and this has been defined in the context section.

Disinvestment is also mentioned in the Introduction but not in the rest of the paper (were participants asked about this?). Since this is a paper about commissioning, I wonder why the term 'disinvestment' was used rather than 'decommissioning'. There is considerable debate in the resource allocation literature about definitions for these terms; sometimes one is used to define the other, sometimes the definitions overlap, sometimes they are mutually exclusive [2]. Although there are a range of definitions, it could be said that 'decommissioning' tends to be used more for funding decisions related to health services and 'disinvestment' used in decisions about monetary and non-monetary resources for removal, reduction or restriction of individual clinical practices within health services. Based on these definitions, decommissioning and disinvestment will have quite different issues and specific barriers. It would be helpful to know what is being referred to in this paper.

The term disinvestment was in common usage whereas decommission was not, at the time and in the context of the study. Definitions of the terms used have been added to the Context section and a reference to the SHARE paper discussing this.

I suggest that appropriate terms are chosen to convey the desired concepts, that the terms selected are defined, and then used consistently within the manuscript.

The terms are now defined in the Context section.

ABSTRACT

Three themes are noted in Results. The first is written with a capital letter. Given the long sentence with lots of punctuation, I think it would help the reader if the first words of the other two themes were also capitalised or a numbering system was introduced (eg 1) methods....., 2) relationships.....etc), or both. While I think the current wording is technically correct, I had to read it twice to pick out the three themes.

This section has been re-written to improve the readability.

INTRODUCTION

For their overview of use of explicit mechanisms for priority setting the authors may wish to know about several other studies that have reported use of decision-making tools for resource allocation – the literature review noted above includes eight (four of which were UK based) [1]. The Australian SHARE Program investigated resource allocation decisions at all levels within a regional health service (ie not just board level) [3, 4] and investigated and applied explicit priority-setting mechanisms [5]. Many of their outcomes are consistent with those reported in this manuscript which would help to strengthen the potential generalisability of the findings.

The authors thank the reviewer for reminding them of the relevance of the SHARE study and this has now been included.

METHODS

Context

The context is well described however, as someone unfamiliar with the UK system, I found it hard to follow some of the details when they were discussed later in the paper. This is a minor point, but it would have helped me to be able to refer to a diagram of the hierarchy for commissioning dental services.

A figure illustrating the structure has now been added (Figure 1)

While I understand that those in the field may be confused about where decisions are made (national or local), and this finding is also consistent with the SHARE program findings [3], I wonder if there is

documentation about who should be responsible for what. Are the local decision-makers unaware of what is done at national level and what is required of them at local level, or is there no guidance on this at all? Does the shortcoming exist at national level (absence of guidance) or local level (lack of awareness of guidance) or is it something else?

The lack of formal guidance and agreement on national versus local split has now been described in the Context section.

I think the acronym RAINDROP should be written in full.

This is now written in full

Recruitment

'...delivered to all NHS England commissioners...' '...including those not present at the seminar'

Another minor point, but if some were not present then the seminar was not delivered to ALL the commissioners. Perhaps it was 'offered to all'?

This wording has been amended as suggested

Sample

There is considerable duplication between Recruitment and Sample. Perhaps combine the subheadings? Or have Sample first to describe all dental commissioners and then Recruitment to note the seminar, personal and email invitations, and follow up?

The sample and recruitment sections have been combined as suggested.

I think it is incorrect to say 'sample' when all members of the relevant group are invited ie there was no sampling from the target group. Perhaps 'participants', 'interviewees' or 'target population'?

"Sample" has been replaced with "Participants"

I think the second half of this paragraph regarding response rate should be in Results.

This section has been moved to results.

Interviews

Were interviewees specifically asked about reallocation and disinvestment as noted above?

'...were aware of the interviewer's background....wider context...' Does this mean the interviewees were made aware of the interviewer's background and wider context at the start of the interview or that they had previous knowledge of this?

This was explained at the seminar and for those not present, at the beginning of the interview. This is now clarified in the manuscript.

I think a more detailed outline of the interview schedule/proforma should be provided. The list of topics does not cover the level of detail about the questions that were asked or prompts used. This raised questions for me as I read through the Results. This could be provided in a supplementary file if the word limit precludes inclusion in the manuscript (although adding it in a table could also get around this).

This has been uploaded as Supplementary File 1 as per editorial request.

I am interested in the reasoning for separating male and female respondents and identifying responses by gender. Identifying subgroups of health service staff is not unusual, but it is usually done because of the potential for different groups to respond differently based on their roles; for example doctors/nurses, clinicians/managers, senior/junior, hospital/community. It may be important to separate responses from male and female patients in clinical trials because of

gender specific factors that influence health, but I am not aware of such factors in health service employment when all participants are in the same role. Was there a hypothesis that male and female commissioners may respond differently? If so, I think it is important that this is documented in the Methods and Results and then considered in the Discussion. What is the evidence to support the hypothesis? How do the results compare to what is known on this topic?

The gender was identified for convention reasons, but the reviewers' point is very valid that there is no evidence to expect any difference by gender and this was not studied. The reference to gender has therefore been removed.

Analysis

After emerging themes were defined '...results were discussed with a sub-sample of participants....'
To what purpose?

Were specific aims for the discussion presented to the sub-sample? What were the aims? Was this process structured?

How did this process influence the results?

This statement was potentially misleading as the results were discussed with some participants to feed them into the priority setting process that is the major element of the wider RAINDROP project. This was not in order to influence the results as they had already been agreed. As this does not influence this manuscript, we have removed this statement to avoid confusion.

RESULTS

I think there should be a subheading for Response. This should include the response/participation rates and details regarding gender if this is considered to be a factor that would influence the responses. How many commissioners were men/women? Does the response rate reflect this ratio? Was there any systematic difference in their responses? What was it? What might that mean? There are 17 quotes, only 5 are male and 3 of these are from the same person. Is this representative of the participation ratio?

A Response sub-section has been added as detailed above. As discussed above, gender has now been removed from the analysis.

Methods of priority setting

As I am unfamiliar with the NHS models, it would have been helpful to have a figure of the commissioning cycle. In my experience 'methods of priority setting' and 'strategic planning' are not the same thing and I would not put the issue of being unable to undertake the whole commissioning cycle or how commissioners undertake strategic planning under this subheading. However my confusion may be due to lack of familiarity with the processes in the commissioning cycle. I

suggest further elaboration on the cycle, and how strategic planning is a method of priority setting in this context, for readers outside the UK.

The commissioning cycle is now detailed in the Methods: Context section, which explains that in the NHS Priority Setting is seen as part of the Strategic Planning phase of commissioning. We have also elaborated in the results to illustrate the priority setting is an aspect of strategic planning.

P8 L23-28. I am confused as to why dental commissioners are covering other providers, could this be explained please. I found the wording ‘...although others saw this as an advantage as they generally had smaller geographic areas...’ to be ambiguous. Does ‘this’ refer to being asked to cover multiple providers ie the advantage is the coverage of multidisciplinary activity? Or is the advantage having smaller geographic areas? Or developing better relationships with providers? From my reading of the current wording, it could be possible that having a smaller geographic area and better relationships with providers are both advantages but that commissioning for multiple providers is still a disadvantage – so it’s not one or the other.

P8 L46. ‘...the increasing geographic area that teams were being asked to cover...’ This seems to be inconsistent with the paragraph above about smaller areas. Perhaps additional elaboration would make this clearer.

In the current structure, some regions have dental specific teams covering several sub-regional areas whereas other regions have generic primary care teams covering single sub-regional areas. We have made this clearer within the results, which hopefully clarifies each of the points you have raised about this section.

P9 L3. Just checking....should ‘nice’ be ‘NICE’?

No, there are no NICE guidelines for orthodontics and upon listening to the audio-recording it is evident that they mean “good”. As a number of the authors also questioned this when we initially analysed the transcripts, I have amended the quote to avoid confusion for readers.

P9 L16-18. ‘...when asked how they would prioritise the areas, no commissioners mentioned either specific systems or tools...’ What are ‘the areas’? I assume this is in response to an open-ended question. Were there any follow up questions asking commissioners if they had used any methods or tools, or any questions that asked them about specific tools? If so, I think this needs to be added. If not, how can the authors be sure that specific methods just didn’t come to mind or that the commissioners didn’t consider the method they used to be a ‘system or tool’? Why is the question phrased to identify future activity (how WOULD they prioritise) rather than reflect past activity (how DID they prioritise)?

The topic guide suggested the question “In your commissioning decisions, what ways do you set priorities for investment/projects?” with further prompts for investments and disinvestments and criteria but as these were semi-structured interviews the specific phrasing and follow up questions would vary depending on the direction of the interview. The text in the manuscript has been amended to clarify that this question was about what they DID and that no commissioners mentioned proactively setting resources (which, as the reviewer points out, is more accurate than saying no tools/systems were used). In addition the “areas” has been clarified.

P9 L44-51. The reference to ‘national direction’ in the text suggests that national direction to use expertise and tools in priority setting would be useful. But the reference to ‘national policy’ in the quote is about ‘continuing’ to do something. If, as previously stated, the commissioners have not been using specific methods and tools then this reference can’t be about national policy to continue to use methods and tools they haven’t been using.

The “national direction” refers to national direction on what the priorities should be and the text has been amended to clarify this.

P9 L48. Just checking....is there a typo or missing words in this quote before or after ‘then’?

The grammar within the quote was wrong and this has been corrected.

P10 L4. If the national contract is held in perpetuity by providers, what local commissioning can occur?

The situations when the contract is adjusted or other resource releasing activities have now been clarified.

P10 L17. The primary care contract is based around UDAs (P4 L35-36) but this quote states 'You can't use a UDA.' These points are confusing, I think further elaboration is required to explain what seems to be inconsistencies.

The quotation has had clarification added to say that the participant meant that the UDA can't be used to allocate resources for priorities and the accompanying text has been adjusted to explain this further.

Relationships with key stakeholders and advisors

P10 L39. 'It seems that this came from three different sources...' I wonder if 'this' should be 'the advice'. The topic of the previous paragraph is the 'need for advice', hence 'this' in the following paragraph would be the need for rather than the actual advice. From my reading of this paragraph I suspect it is the latter.

"This" has now been clarified to "The advice"

The findings are reported as 'confusion and disagreement' but only one quote is given. The current quote does not convey confusion and at least two are required to demonstrate disagreement.

Confusion and disagreement has been changed to "lack of clarity" and an extra quote has been added to support this.

P11 L5. What does 'level of relationship' mean? Should 'depending' be 'depended'?

This has been clarified to "strength of relationship" and "depending" has been corrected.

P11 L23-26. Why do commissioning teams employ dental advisors for a purpose other than commissioning? And why is this a problem for the commissioning process? There is no quote for this finding.

The role of DPAs (quality management and contract monitoring) has been outlined in the text and a quote added.

P11 L30-31. How do good relationships with LDNs and CDPHs create better opportunities for priority setting, resource allocation and commissioning in general? I'm not sure what 'opportunities' means in this sentence or how opportunities are affected by these relationships. There is no quote for this finding.

This has been clarified to "felt more able to undertake" and a quote has been added.

P11 L36-39. What does 'limited' mean here – absent, insufficient, poor quality, something else? Why are non-dental commissioners stakeholders?

Limited was used here to mean all of these things, but they have now been spelt out. Non-dental commissioners are stakeholders as there are many areas of overlap between dental and general health and health services (e.g. emergency and out of hours care, health promotion, care of special needs groups). This has been explained.

National versus local responsibility

This is referred to as 'Tensions between national and local responsibility' in the abstract. Should it be 'responsibilities'? I think the theme should be written the same way in both places.

"Responsibilities" has now been used throughout the manuscript.

As noted earlier, I think more elaboration about the process is required. Is there an established national process that local commissioners are unaware of, or is it ambiguous, or confusing, or absent, or something else? There is a suggestion in the quote that it might be 'devolving' – is it? Is this the cause of the confusion? It's hard to tell what are facts that could be substantiated (eg that there is/isn't clear documentation on the responsibilities at national/local level, or that the national

process is/isn't to be devolved to local bodies) and what are the perceptions of local commissioners (which could be right or wrong).

Clarification has been added to the Context section. However, anecdotally, from the authors personal experience, there is no documentation or actual decision about much of this, with different answers to questions about power and responsibility given by different senior figures or at different times.

P12 L42-48. This seems to be a good quote to illustrate national/local confusion but not to illustrate limited flexibility or delays in central decisions.

This quote has been moved to align with national/local confusion and a new quote has been added to illustrate delays

P12 L53-55 and P13 L8. Are the 'dental leads' dental commissioners? Are the 'local teams' the 'dental care teams'? What is the 'hierarchy locally'? What is the relationship between dental commissioners and dental care teams? Who are the 'Directors'? A figure with these relationships would be really helpful to those unfamiliar with NHS structures.

A figure has been added to the Context section and these terms have been standardised in the Results section to match with those introduced in the Context section.

P12 L15-21. Just checking....is there a typo or missing words/punctuation in this quote? The words '...the structure we were set up either give us the autonomy....' don't make sense to me.

This quote has been clarified.

DISCUSSION

Several claims are made that I could not find in the Results section.

'The ...study ...found ...commissioners felt that resource re-allocation was needed but did not happen often or systematically.'

This statement has been amended to "felt that there were problems with current resource allocation but that proactive, systematic priority setting for the allocations did not happen" reflecting the results on pTO DO

'Where links were strong, more priority setting seemed to occur.'

This statement has been amended to "Where links were strong, commissioners felt more able to undertake priority setting" reflecting the results on pTO DO

'...tension between the national team and the regional commissioners.' (There is no mention of the national 'team' only responsibilities)

The revisions to the results now make clear there is a national team.

□ ‘...frustration with national direction not aligning with local needs...’

This statement has been amended to “frustration with a lack of national direction” reflecting the results on pTO DO and also the findings of the cited study.

□ ‘...good links with clinical advice should be prioritised and grown...’

This statement has been added “given that commissioners felt more able to set priorities when good links were in place” to illustrate how this recommendation relates to the results.

P13 L51-53. ‘...with those actually making resource allocation decision rather than at a higher managerial level...’ The interviewees were dental commissioners. What would be the higher managerial level of people involved in commissioning dental services? Why would they be interviewed rather than the dental commissioners?

This refers to the national dental team and also sub-regional board/directors. The statement has been amended to clarify this and the role they may have in priority setting and resource allocation. “As described in the introduction, the study is unusual in conducting in-depth interviews with those actually making resource allocation decisions (i.e. sub-regional commissioners) rather than at a higher managerial level (e.g. board members, financial directors, or national leaders, who may also need to undertake priority setting and resource allocation decisions but at a more strategic level)...”

P14 L3. The wording ‘...reasons underlying resource allocation could be explored in depth’ is ambiguous. Reasons for what? Is this referring to resource allocation decisions, the resource allocation process, or something else? No criteria used in resource allocation decisions are mentioned. Some barriers to use of specific tools for prioritisation are mentioned, but the process of resource allocation [3], or any reasons related to how resource allocation is practised are also not mentioned.

This has been clarified to “the reasons for adopting specific approaches to resource allocation”

P14 L10-13. While there are still some limitations to the generalisability of this study, some general findings (lack of time, skills, etc) are found in many other studies [1] and more specific findings (confusion about who makes final decisions, lack of use of prioritisation tools) were also found in the SHARE studies [3, 5] and others [1], which strengthens the findings herein. If it was established that national direction does not align with local needs, this has also been reported by others [6].

The discussion now includes these references as examples of similar findings.

P14 L35. ‘...markedly different structures...’ Different to what? How are they different? How would you expect the differences to impact on the findings?

The previous structure is now described (briefly) to illustrate the former weaker link between national and local decision making/advice and increased autonomy of local bodies.

P14 L 43. Why are PBMA, MCDA, A4R and STAR recommended when Public Health England has developed a specific tool for this purpose?

The PHE tool is intended to be used for the commissioning that PHE is directly responsible (i.e. public health interventions) and NHS England is a separate body altogether. This has been clarified in the Introduction and PHE’s responsibility is also illustrated in Figure 1.

CONCLUSION

P15 L12. '[Lack of] time and expertise were barriers...'

This has been corrected.

Reviewer: 2

Reviewer Name: Martina Garau

Thanks for the opportunity to review this article which explores and provide evidence on the important topic of lack of systematic and efficient priority setting for the provision of part of NHS services. The focus on dentistry is well justified and represent a useful example of disconnect between national and local decision making and call for a need for more explicit resource allocation processes.

The method is well explained and appropriate for the research question. The discussion provides a good comparison of the results with previous literature.

The article, however, introduces a number of concepts and terminology (probably used by some of the interviewees) that needs further explanation and analysis. Given the variety of expertise and nationality of BMJ readers, it would be useful to clarify what is meant by "explicit" priority setting processes and tools, and to explain what the role of NICE guidelines is in decision making, for example.

The Introduction and Context sections now include more detailed explanations of the concepts and terminology.

The themes identified are relevant and they provide valuable evidence of the existing issues. However, more analysis or interpretation of interviewees' quotes would improve the quality of the article.

Many of the quotes are now interpreted further and explained in the context of the study.

Finally, the article provides some clear messages but many sentences are not easy to follow and understand. Therefore I suggest re-drafting part of the text to increase clarity and readability. Some of the sentences are pointed out in my specific comments below.

Many sentences have been re-worked to simplify them. Specific examples below are dealt with in turn below.

Specific comments:

The title should specify that the study is focused on dentistry services in England

The title has been revised to include reference to dentistry in England

The text in the introduction is cryptic in parts and might benefit from some more specific examples and recent references in the literature, for example:

Page 2, line 26: sentence might need re-drafting as not very clear ("where the variable but key nature of relationship with clinical advice sources was revealed")

This has been re-written.

Page 3 line 19: the sentence with "prioritisation was undertaken" should specify the setting and the timing the authors refers to

The setting and time are now referred to in this statement.

Page 3 line 26: it might be useful to specify what “traditional economic evaluation” techniques are (with a more recent/better known paper referenced) and maybe refer to their use within the NICE programmes, given that the audience might include non-economists

This section has now been re-written to go beyond health economics approaches at the request of reviewer 1 and so this is no longer relevant.

Page 3 line 50: Karlsberg Schaffer et al. also published a study on decision making in Scottish NHS boards (Karlsberg Schaffer et al. (2015) Local health care expenditure plans and their opportunity costs, Health policy Volume 119, Issue 9, Pages 1237-1244). More generally, unless a literature review was conducted, it might be better to soften the tone of the sentence in line 50 (“two notable exceptions”).

The tone in this sentence has been softened and the authors thank the review for this reference which is now incorporated.

Page 4 line 15: it is not totally clear the difference between board and decision making level. In the Karlsberg Schaffer et al. articles board is actually meant to represent the decision making level so need further explanation.

“Board” has been replaced with “director” to clarify this.

Page 4 line 32: “is” should be “are”

Corrected

Page 4 line 28: is it possible to provide an example of “sub-region?”

An example is now given

Given that the title allude to a possible tension between national and local decision making, the issue should be better presented in either the introduction or the context.

This is now explored further in the Context section

Page 5 lines 17 and 34: here the word “regional areas” is used while on page 4 line 28 the word “sub-regional” is used

“regional areas” has been corrected to “sub-regional”

Page 7 line 24: is there a reason to indicate gender in the quotes?

As described in the response to reviewer 1, we would agree that there is no need, and these have been removed.

Page 7 line 43: what is the “contracting element”? and “commissioning skills”? If these concepts/wording were used by interviewees, there is a need to explain what they are to help the readers

The commissioning cycle is now explained in the Context and “contracting element” has been defined. Commissioning skills is also further defined.

Page 10 line 10: what does “GDS contract” stand for?

This relates to the General Dental Services contract, but is now explained in the text.

Page 10: consistency is needed in the use of Local Dental Network, LDT, and LPN

This has been revised throughout the paper

Page 13 line 42: the sentence in bracket explaining what explicit processes are is key so should not be in bracket and further expanded

This is now defined in the introduction but the definition has been repeated in the discussion as suggested.

Page 13 line 31: through the text the authors use the words “allocating” and “re-allocating” but it is unclear if there is a difference between the terms

To simplify interpretation, allocating has been used throughout the article.

VERSION 2 – REVIEW

REVIEWER	Claire Harris Monash University, Australia
REVIEW RETURNED	12-Nov-2018

GENERAL COMMENTS	Thank you for asking me to re-review this paper. The authors are to be commended, the additional information provided in tables and figures, as well as the amended section on context with consistent use of terminology, makes the paper much easier to read (for those of us less familiar with details of the NHS structure and processes) and strengthens the discussion and conclusions.
--